# Effect of sensor number and location on accelerometry-based vertical ground reaction force estimation during walking

**Ricky Pimentel** [1], **Cortney Armitano-Lago** [2], **Ryan MacPherson** [2], **Anoop Sathyan** [3],
**Jack Twiddy** [1], **Kaila Peterson** [4], **Michael Daniele** [1,4], **Adam W. Kiefer** [2], **Edgar Lobaton** [4],
**Brian Pietrosimone** [2], **Jason R. Franz** [1]*

1 Joint Department of Biomedical Engineering, University of North Carolina at Chapel Hill & North Carolina State University, Chapel Hill & Raleigh, North Carolina, United States of America, 2 Department of Exercise & Sport Science, University of North Carolina at Chapel Hill, Chapel Hill, North Carolina, United States of America, 3 Department of Aerospace Engineering, University of Cincinnati, Cincinnati, OH, United States of America, 4 Department of Electrical and Computer Engineering, North Carolina State University, Raleigh, North Carolina, United States of America

* jrfranz@email.unc.edu

**Data Availability Statement:** The data that support the findings of this study are publicly available from

## Abstract

Knee osteoarthritis is a major cause of global disability and is a major cost for the healthcare system. Lower extremity loading is a determinant of knee osteoarthritis onset and progression; however, technology that assists rehabilitative clinicians in optimizing key metrics of lower extremity loading is significantly limited. The peak vertical component of the ground reaction force (vGRF) in the first 50% of stance is highly associated with biological and patient-reported outcomes linked to knee osteoarthritis symptoms. Monitoring and maintaining typical vGRF profiles may support healthy gait biomechanics and joint tissue loading to prevent the onset and progression of knee osteoarthritis. Yet, the optimal number of sensors and sensor placements for predicting accurate vGRF from accelerometry remains unknown. Our goals were to: 1) determine how many sensors and what sensor locations yielded the most accurate vGRF loading peak estimates during walking; and 2) characterize how prescribing different loading conditions affected vGRF loading peak estimates. We asked 20 young adult participants to wear 5 accelerometers on their waist, shanks, and feet and walk on a force-instrumented treadmill during control and targeted biofeedback conditions prompting 5% underloading and overloading vGRFs. We trained and tested machine learning models to estimate vGRF from the various sensor accelerometer inputs and identified which combinations were most accurate. We found that a neural network using one accelerometer at the waist yielded the most accurate loading peak vGRF estimates during walking, with average errors of 4.4% body weight. The waist-only configuration was able to distinguish between control and overloading conditions prescribed using biofeedback, matching measured vGRF outcomes. Including foot or shank acceleration signals in the model reduced accuracy, particularly for the overloading condition. Our results suggest that a system designed to monitor changes in walking vGRF or to deploy targeted biofeedback may only need a single accelerometer located at the waist for healthy participants.

DBHub.io with the identifier(s) [https://dbhub.io/peruvianox/Pilot1_steps.db].

**Funding:** JRF and BP received funding from the UNC Chapel Hill Eshelman Institute for Innovation for this study. The funders had no role in study design, data collection and analysis, decision to publish, or preparation of the manuscript.

**Competing interests:** Jason R. Franz and Brian Pietrosimone are founders and Chief Technology Officer and President, respectively, of VETTA Solutions, Inc, a start-up company with prospective product solutions in the areas of wearable sensor solutions for musculoskeletal health applications. The conceptualization of this study and all data collections and analyses were completed prior to the incorporation date of the company.

## Author summary

Knee osteoarthritis is a major cause of global disability and a major cost for the healthcare system. Lower extremity loading is a determinant of knee osteoarthritis onset and progression; however, technology that assists rehabilitative clinicians in optimizing key metrics of lower extremity loading is significantly limited. Monitoring and maintaining typical limb loading profiles may support healthy gait biomechanics and joint tissue loading to prevent the onset and progression of knee osteoarthritis. Yet, the optimal number of sensors and sensor placements for predicting accurate limb loading from accelerometry remains unknown. Our goals were to: 1) determine how many sensors and what sensor locations yielded the most accurate limb loading peak estimates during walking; and 2) characterize how prescribing different loading conditions affected limb loading peak estimates. We found that a neural network using one accelerometer at the waist yielded the most accurate loading peak estimates during walking, with average errors of 4.4% body weight. Our results suggest that a system designed to monitor changes in limb loading during walking or to deploy targeted biofeedback may only need a single accelerometer located at the waist for healthy participants.

## Introduction

Osteoarthritis (OA) is the 10th leading cause of global disability, with approximately 10% of adults in the United States exhibiting OA [1]. In addition to substantial physical burden, management of OA exacts a considerable financial burden on our healthcare system.[2] As of 2019 in the United States, OA annually costs approximately $360 billion in direct costs and $550 billion in all-cause total costs [1]. Although OA risk increases with age, the consequences of OA are not limited to older adults. Individuals who sustain lower hip, knee, or ankle joint injuries are at much higher risk for developing OA early in life [3]. Considerable effort and investment has led to surgical and rehabilitative interventions to prevent the development and slow the progression of OA. Aberrant loading is a key determinant of OA onset and progression [2–8]. Unfortunately, most interventions are not specifically designed to optimize loading of lower extremity joints, which is critical to maintaining healthy joint tissues [9–11].

Healthy gait biomechanics distribute and balance forces across lower extremity joint surfaces to preserve joint function and tissue health [12,13]. Joint injury and surgery are accompanied by pain, swelling, and muscle weakness which causes individuals to adopt atypical patterns of force distribution across lower extremity joints well after being discharged from formal physical therapy that seeks to correct these impairments [14–16]. Although such gait adaptations may allow individuals to maintain function following injury and surgery, joint tissues break down quickly in response to atypical and unbalanced joint forces [3,5,7,8]. Therefore, it is critical to maintain characteristically healthy gait biomechanics and joint tissue loading to prevent the development and/or progression of lower extremity OA.

Fortunately, limb-level biomechanical outcomes are linked to joint tissue and symptom-level changes associated with OA onset and progression [8,17–20]. Specifically, lower first peak vertical ground reaction force (vGRF), in particular, associates with more deleterious biological joint tissue changes and worse patient-reported outcomes that are consistent with knee osteoarthritis development [8,17–20]. Altering peak vGRF magnitudes via real-time biofeedback may normalize limb-level gait biomechanics [21] and limit the biological changes leading to OA development. However, personalized prescription of joint loading cannot be feasibly

implemented in the clinical management of patients with OA because of the expensive and often immobile force-sensing equipment required. Thus, there is a critical need for inexpensive and portable systems that can monitor and prescribe evidence-based changes to critical variable of limb-level loading for the purpose of mitigating OA onset and progression.

Cost-effective wearable sensor solutions may provide a clinically-feasible option to monitor limb loading. For example, accelerometers and inertial measurement units have been used to estimate vGRFs during walking. Veras et al. (2022) found that a hip-worn accelerometer narrowly outperformed other accelerometers (distal shank, lower back) in estimating vGRF loading peak during walking (hip: $R^2 > 0.96$, mean absolute percentage error <7.3%) [22]. Similarly, Alcantara et al. (2021) estimated loading peak vGRFs during running using a sacral accelerometer within a mean absolute error (MAE) of 4.3%BW (percent body weight) using a quantile regression forest model and of 4.0%BW using a linear regression model [23]. Even more researchers have used accelerometers below the waist to estimate vGRF. Jiang et al. (2020) found a shank accelerometer to be the single best estimator of vGRF (within a root mean square error of 2%) compared to other sensors on the foot, distal thigh, and proximal thigh. However, those authors did not include comparisons to a waist- or hip-worn sensor [24]. Bach et al. (2022) used only shank accelerometers to estimate vGRF profiles with $R^2 = 0.97$ and a normalized root mean square error of 5.2% [25]. Altogether, these studies highlight that various accelerometer numbers and locations can be used to estimate vGRF. However, none of these studies actively prescribed vGRF changes to determine veracity in predicting effects of loading interventions or to drive those interventions directly.

Based on the available literature, researchers and clinicians seeking to estimate walking vGRF via accelerometry may be confused regarding where to place sensors and how many are needed for accurate outcomes. Thus, our first goal was to determine how many sensors and which locations yielded the most accurate vGRF loading peak estimates. Those individuals may also wonder whether wearable devices can reliably detect changes in loading profiles during walking. Accordingly, our second goal was to characterize how different loading conditions prescribed using biofeedback affect vGRF loading peak estimates. Ultimately, this is an important step in developing wearable sensor and biofeedback systems that can be deployed in real-world environments to detect, monitor, and treat aberrant forces associated with lower extremity OA.

## Materials and methods

### Ethics statement

The University of North Carolina at Chapel Hill Institutional Review Board approved this study (#21–1693) and all participants provided informed consent prior to participating in any study activities.

### Participants, equipment, & experimental design

We recruited a cohort of 20 (10 male, 10 female) healthy young adults to participate in this study (age: 24.7±5.18 years, height: 1.77±0.11 m, mass: 75.6±13.7 kg, typical walking speed: 1.41±0.09 m/s). We excluded any prospective participants who: were younger than 18 or older than 35 years of age; had a history of congenital or acquired cognitive, ophthalmologic, or neurological disorders; had a history of chronic peripheral or central vestibular disorders; began anti-seizure medication for any reason within two months of participation; had a BMI equal to or greater than 36; or used an assistive device and/or orthotics to walk. We measured participants' habitual overground walking speed via four passes along a 10-m walkway. We used Delsys Trigno (Natick, MA, USA) sensors not to collect electromyographic recordings, but to

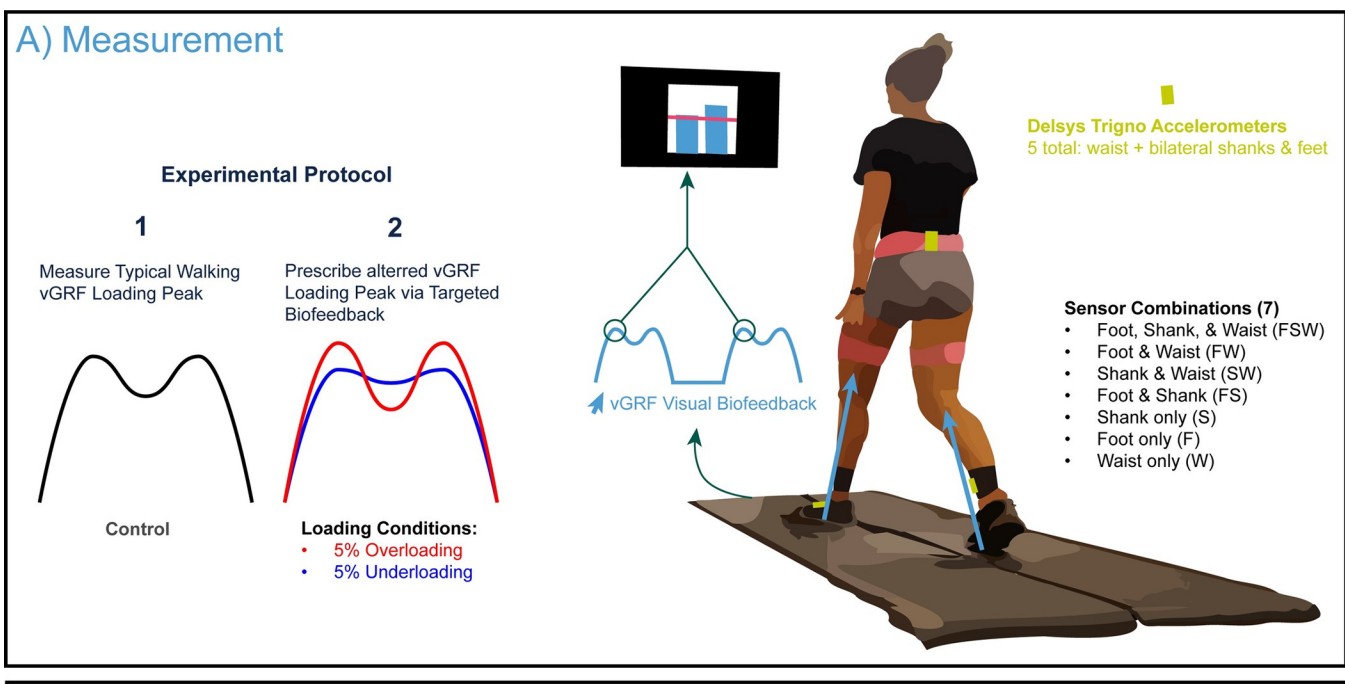

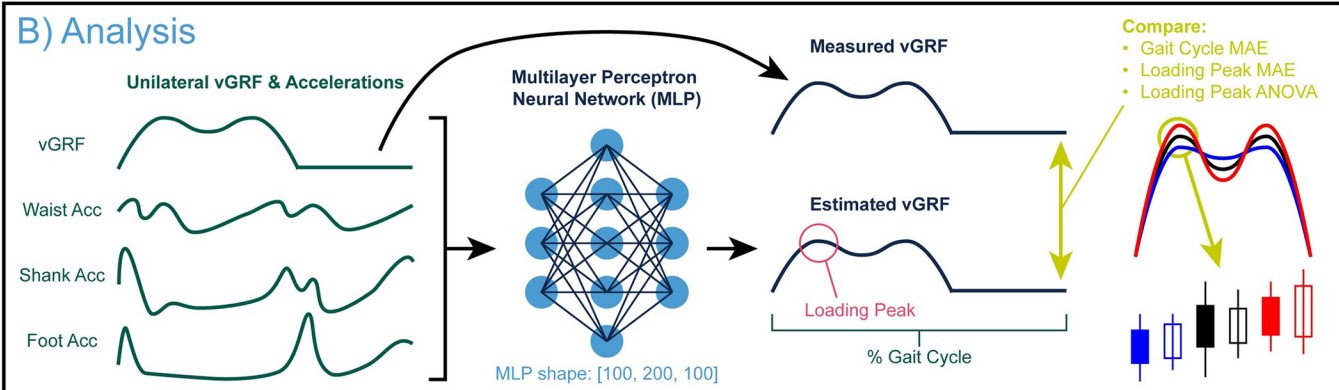

**Fig 1.** A) We recorded ground reaction forces and lower body accelerations during typical walking and prescribed changes using vGRF targeted biofeedback displaying the loading peak on a screen in front of the participant. B) For analysis, we trained a series of MultiLayer Perceptron models to estimate the unilateral vertical ground reaction force (vGRF) across the gait cycle based on acceleration inputs from various sensor combinations that included waist, shank, and foot sensors. All trained models unilaterally estimated vGRF, only including sensors on that specific side (i.e., left vGRF from the waist, left shank, and left foot).

collect bilateral three-dimensional acceleration from the dorsum of each foot, the anterior/medial bony surface of the distal shank, and posteriorly on the waist (Fig 1A). The version of our Trigno sensors did not include gyroscope or magnetometer measurements.

Participants walked at their typical overground speed on a force-instrumented treadmill (Bertec, Columbus, OH, USA) for five minutes (control condition, Fig 1A). We recorded synchronized acceleration (2000 Hz) and GRF data (1200 Hz) via Qualisys Track Manager (Qualisys, Gothenburg, Sweden). During the third minute of this control condition, we also recorded vGRF profiles using custom Matlab (Mathworks, Natick, MA, USA) scripts [26] to use for subsequent trials with targeted biofeedback (Fig 1A). Specifically, participants performed two additional five-minute walking trials while responding to real-time biofeedback driven by instantaneous force measurements from the instrumented treadmill. Visual

biofeedback was designed to prescribe, in randomized order, 5% higher and 5% lower loading peak vGRF (overloading and underloading, respectively) during the first 50% of the stance phase (Fig 1A). Participants viewed a screen showing two bar plots, representing their left and right vGRF loading peak, updated in real time as the average from the two previous steps (Fig 1A). As displayed in Fig 1A, we instructed participants to match their peak vGRF (blue bars updated on every step) with the target force (horizontal red line) to the best of their ability. The ±5% target values for these loading conditions were based on the side-specific averages measured during the control condition and were designed to emulate different lower extremity loading phenotypes.

To facilitate sensor synchronization, we asked participants to stomp on each force plate with their left and right foot in succession prior to starting or stopping the recorded data. The pre/post-trial stomp provided a sufficient timestamp via a high magnitude signal for a reliable cross correlation analysis. Out of the 60 total trials recorded, two trials needed to be manually adjusted due to a synchronization error between the acceleration and force data. For these two trials, we synchronized the acceleration and force data by identifying the relative lag using a cross-correlation function and truncating or appending empty frames to the acceleration signal, keeping the force signal constant.

## Data reduction & model descriptions

We extracted the first 50 valid left and right strides for a total of 100 strides per walking condition. We resampled acceleration and GRF signals to 100 Hz in post processing. For consistency across both vGRF and acceleration signals, we extracted gait cycles using a peak-finding algorithm force data normalized to percent body weight (%BW), identifying stance phases with a minimum height of 95%BW, a minimum prominence of 80%BW, a minimum width of 0.15 s, and a minimum distance of 0.5 s between peaks. We selected these identification parameters based on manual observation across the entire dataset (all subjects and biofeedback conditions).

We extracted accelerometer signals in time series for foot, shank, and waist with time-matched vGRF data and resampled each stride to 100 data points (Fig 1B). To remove any bias of sensor orientation and sidedness, we calculated the acceleration vector magnitude (or Euclidian norm) as: $Acceleration\ Vector\ Magnitude = \sqrt[2]{x^2 + y^2 + z^2}$, where x, y, and z, are the 3D orthogonal components of the acceleration measurement. Fig 2 shows the measured vGRF and acceleration data for all input steps.

We tested seven different sensor combinations by altering the number and location of accelerometers used as model inputs, as displayed in Fig 1A. Each model included steps across all loading conditions (control/under/over). We trained Multi-Layer Perceptron regressor models from the Sci-Kit Learn [27], with a convergence max iteration of 500, logistic activation functions, and a single hidden layer (size = 200 neurons) based on preliminary parameter tuning (Fig 1B). We trained the MLP neural networks to estimate the ground truth vGRF signal recorded via instrumented treadmill as a 100-point vector using a Nx100-point accelerometer input corresponding to the sensor(s) used (1x100 for a single sensor, 2x100 for two sensors, etc.). These 100-point signals represent the vGRF and accelerometer recordings interpolated to percentage of the gait cycle (%GC), measured between consecutive heel strikes during walking. To benchmark model accuracies, we performed 5-fold cross validation, splitting the training and testing steps on an 80:20% (16:4 subjects) ratio in a subject leave-out approach. In line with common k-fold cross validation practices, we changed which 20% the model was tested on for each k-fold iteration.

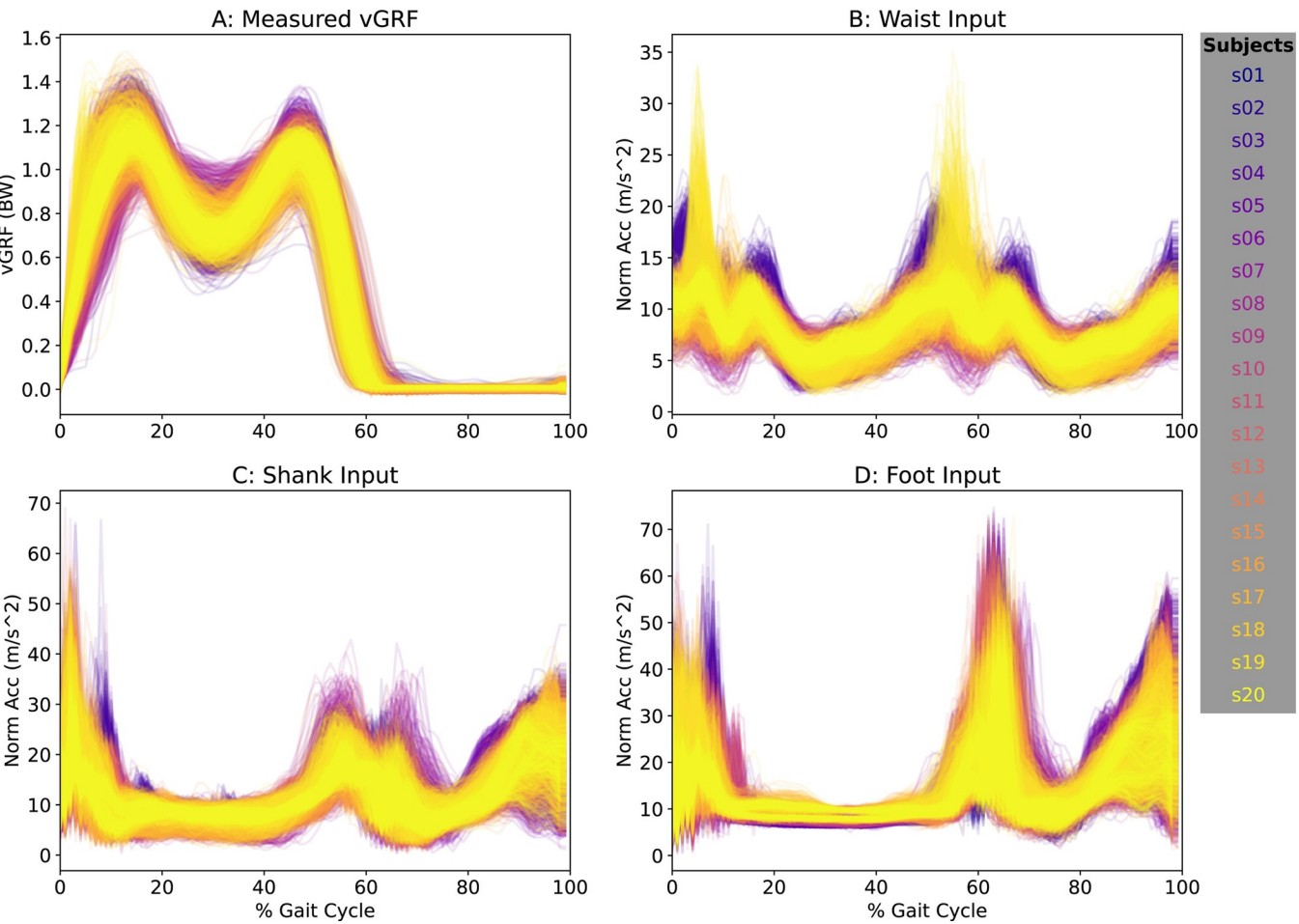

**Fig 2.** Here we show the target vGRF signal (A) and input vector magnitude acceleration waveforms (B-D) used to estimate vGRF for 100 steps (50 left, 50 right) for all subjects (separated by color).

### Statistical comparisons

We quantified accuracy via mean absolute error (MAE) between measured and predicted vGRF both across the entire gait cycle (GC) and at the loading peak. We characterized the ability to distinguish between loading conditions by performing two-way repeated measures analysis of variance (ANOVA) using each of the three most-accurate models, testing for main effects of mode (measured vs estimated), condition (under, control, & over-loading), and for interaction effects (mode x condition). We also ran one-way repeated measures ANOVA across the MAE estimates across the GC and at the loading peak. When a significant main effect was found, Tukey's post-hoc tests identified significant pairwise differences between measurement modes and across loading conditions. We report effect sizes for ANOVAs as partial eta squared ($\eta_p^2$) and post-hoc analyses using eta squared ($\eta^2$).

## Results

### Benchmarking accuracy across sensor configurations

Fig 3 shows all measured (panel A) and predicted vGRFs for all steps in the testing set across each sensor configuration (panels B-H) and k-fold iteration (line colors). Qualitatively, all

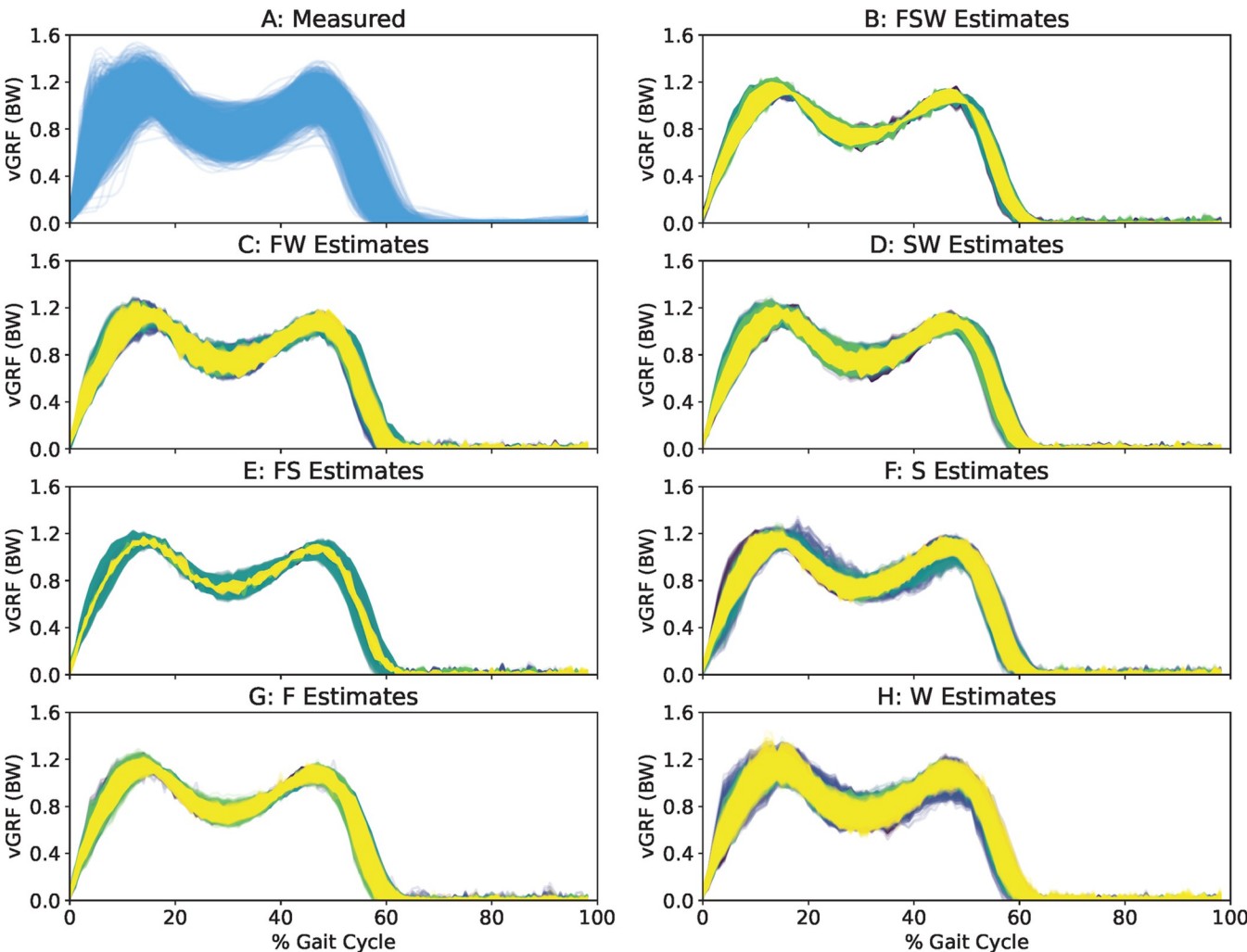

**Fig 3.** For qualitative comparison convenience, we show the measured vGRF signal (A) and all model estimates (B-H) for all steps across each k-fold validation (color). All models successfully estimated the double-humped vGRF profile across the gait cycle. The Waist model seemed to best reflect the true variability of the original signal.

sensor configurations yielded model predictions that reflected the double-hump shape of vGRF across the gait cycle. Compared to other models, the Waist model (Fig 3H) appeared to best reflect the variability of measured vGRFs across participants and conditions (Fig 3A).

Fig 4 shows MAE between measured and estimated vGRFs over the gait cycle (A) and at the instant of the loading peak (B) for each sensor configuration (x-axis) and k-fold iteration (colors). The model driven exclusively by Waist accelerations outperformed all other models (GC MAE: 4.0%BW, loading peak MAE: 4.4%BW). The Foot-Waist configuration performed second best (GC MAE: 4.3%BW, loading peak MAE: 5.0%BW) followed by the Shank-Waist configuration (GC MAE: 4.5%BW, loading peak MAE: 5.6%BW). All other sensor configurations produced estimates higher than these reported MAE values, and generally above 5%BW.

## Model-predictions of altered lower-extremity loading

Compared to the control condition, the measured and average estimated vGRF loading peaks increased and decreased consistently with the biofeedback target values (condition main effect:

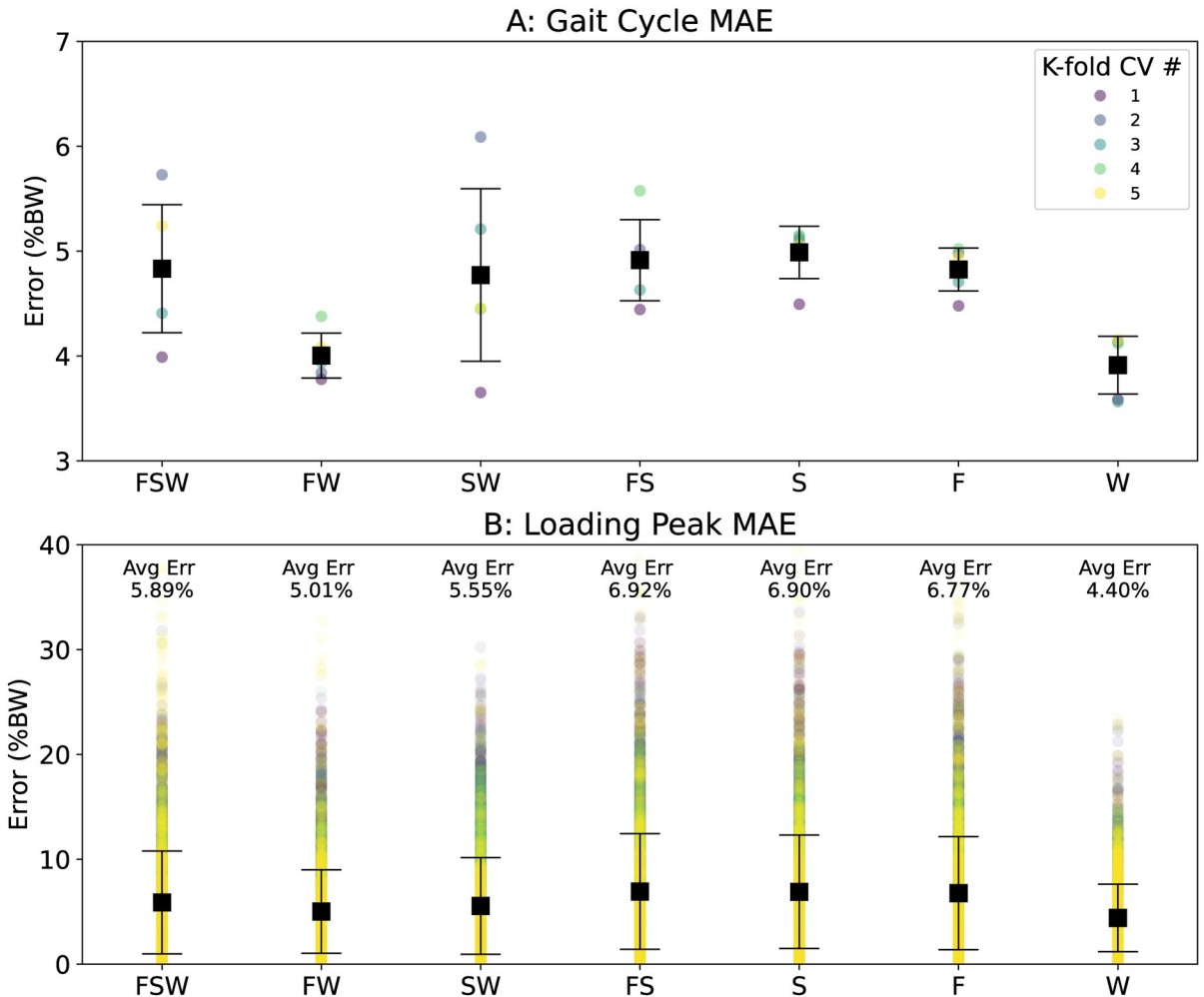

**Fig 4. Model accuracy varied across each of the sensor combinations yet was relatively stable across each k-fold iteration.** The Waist sensor configuration yielded the most accurate model on average for both the gait cycle MAE (A) and loading peak MAE (B). The Foot-Waist model was 2nd best, and Shank-Waist model 3rd best across both outcomes. In panel A we plot the k-fold average in colored circles for simplicity, however in panel B we plot estimates for all steps in circles colored by k-fold iteration. The black squares and bars show the mean and standard deviation in both panels A and B.

p<0.001, $\eta_p^2$ = 0.719). The measured vGRF loading peaks significantly differed from control only for the overloading condition (Tukey's: p = 0.009, $\eta^2$ = 0.177, Fig 5C, 5G and 5K).

The Waist model successfully distinguished between overloading and control vGRF loading peaks (condition: p<0.001, $\eta_p^2$ = 0.705) and yielded values that were indistinguishable from those measured (mode: p = 0.067, $\eta_p^2$ = 0.166, Fig 5C). Waist model MAEs did not significantly change between conditions across the gait cycle (condition: p = 0.460, $\eta_p^2$ = 0.040, Fig 5B) nor at the loading peak (condition: p = 0.087, $\eta_p^2$ = 0.121, Fig 5D).

Compared to the control condition, the Foot-Waist model (condition: p<0.001, $\eta_p^2$ = 0.715) did not distinguish loading peaks for overloading (p = 0.094, $\eta^2$ = 0.100), but did yield a significant difference for underloading (p = 0.027, $\eta^2$ = 0.160, Fig 5G). Compared to measured values, the Foot-Waist model yielded a different loading peak vGRF for overloading (p = 0.018, $\eta^2$ = 0.133, Fig 5G) but not underloading or control (p>0.300, $\eta^2$<0.027). The Foot-Waist model also yielded similar estimates of the vGRF across the gait cycle (condition: p = 0.386, $\eta_p^2$ = 0.049, Fig 5F). Conversely, Foot-Waist model accuracy at the loading peak

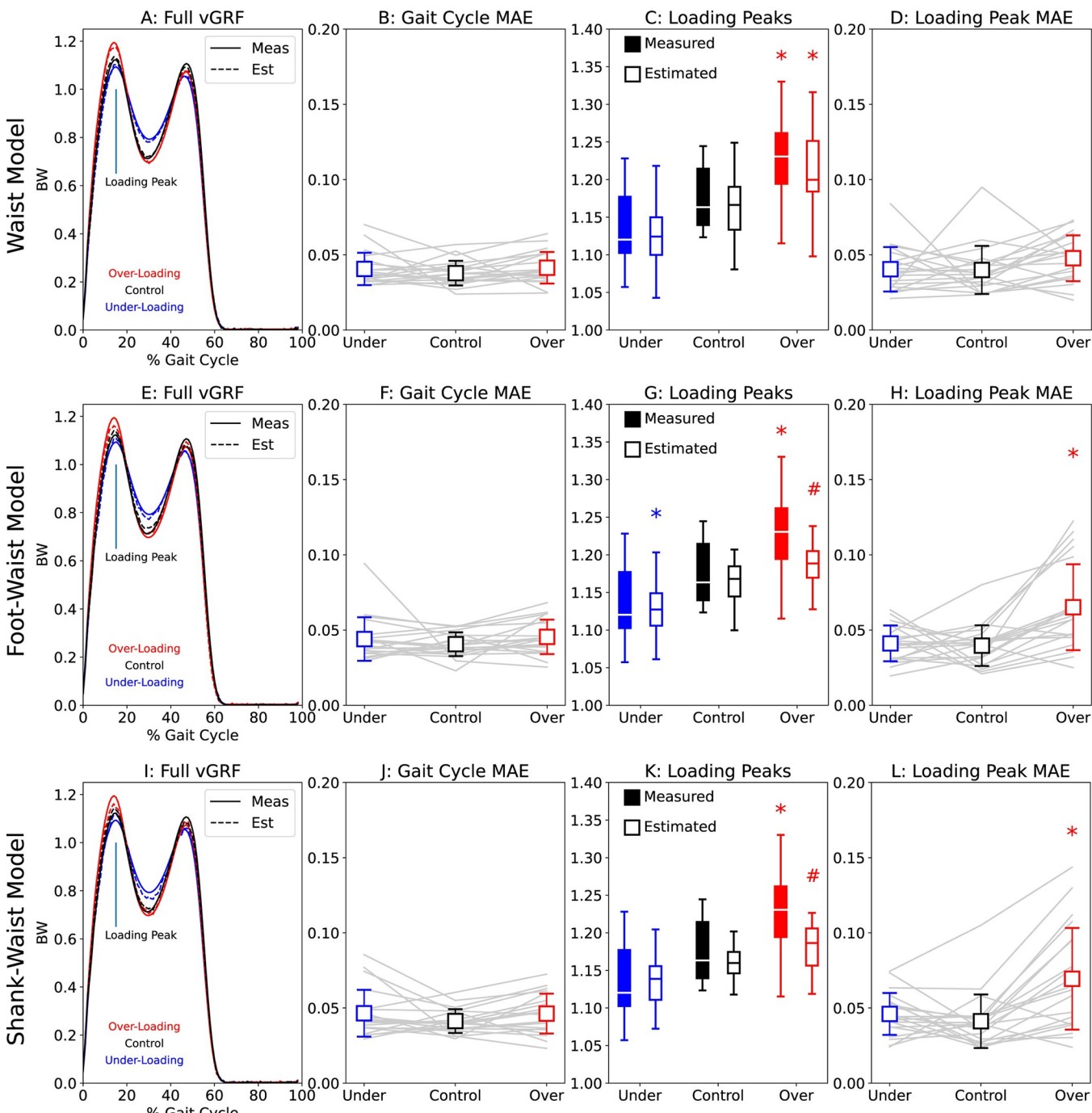

**Fig 5.** We further explore the three most accurate sensor configurations, analyzing the Waist (A-D), Foot-Waist (E-H), and Shank-Waist (I-L) models across each of the loading conditions (underloading = blue, control = black, overloading = red). Asterisks indicate a significant difference from the control condition for that measurement type, whereas hashes (#) represent a significant difference between estimated (open boxes) and measured (filled boxes) vGRF loading peak.

varied across conditions (condition: $p<0.001$, $\eta_p^2 = 0.361$, Fig 5H) with overloading error significantly higher than for underloading and control conditions ($p\leq0.011$, $\eta^2\geq0.147$).

The Shank-Waist model (condition: $p<0.001$, $\eta_p^2 = 0.795$) was unable to distinguish between loading conditions ($p>0.103$, $\eta^2<0.096$, Fig 5K). Additionally, the Shank-Waist

model estimates were different from the measured vGRF loading peaks (mode: $p = 0.032$, $\eta^2 = 0.219$), with the overloading condition estimates different from measured values ($p = 0.003$, $\eta^2 = 0.208$, Fig 5K). The Shank-Waist model yielded estimates with consistent errors between conditions across the gait cycle (condition: $p = 0.051$, $\eta_p^2 = 0.145$, Fig 5J). However, the Shank-Waist model errors at the loading peak varied between conditions (condition: $p<0.001$, $\eta_p^2 = 0.434$, Fig 5L) with overloading error significantly higher than underloading and control conditions ($p \leq 0.015$, $\eta^2 \geq 0.134$).

## Seeking interpretability

Fig 6 explores characteristics of the multilayer perceptron model for the Waist-only sensor configuration. We show a heatmap of the model coefficient weights for the input (waist acceleration or $W_{Acc}$, vertical axis) and output (vGRF, horizontal axis). In the sub-plots, we also show the mean-normalized average weights across each instance of the gait cycle (black bars) and the average input $W_{Acc}$ and output vGRF signals (gray line).

The estimated vGRF (horizontal axis and top sub-axis) demonstrated strong positive weights during early (5–15% GC) and late (50–60% GC) stance phase, as well as strong negative weights during leg swing (60–100% GC). Although the vGRF weights during midstance (15–50% GC) included both positive and negative values, those weights were relatively smaller. Model weights for the input $W_{Acc}$ vectors (vertical axis and right sub-axis) were more complicated. However, we tended to see more positive weights during the stance phase (0–60% GC) and more negative weights during the swing phase (60–100% GC).

## Discussion

Our goals were to: 1) determine how many sensors and what sensor placement and locations yielded the most accurate vGRF loading peak estimates during walking; and 2) characterize how prescribing different loading conditions affected vGRF loading peak estimates. We found that a neural network using one waist-mounted accelerometer yielded the most accurate loading peak vGRF estimates during walking, with an average error of 4.4% BW. Furthermore, this waist-only configuration was able to distinguish between control and overloading conditions prescribed using biofeedback, matching measured vGRF outcomes. Including foot or shank acceleration signals in the model reduced accuracy, particularly for the overloading condition. Our results suggest that a system designed to monitor changes in walking vGRF or to deploy targeted biofeedback may only need a single accelerometer located at the waist for healthy participants.

The novelty in our scientific contributions is twofold. First, we systematically compared model-prediction accuracies across multiple sensor configurations. Previous studies designed to predict walking vGRF values from wearable sensors have generally quantified model accuracies based on a single accelerometer placed at, for example, the feet versus shank or waist. By using multiple combinations of sensors, we see the effects of how the various sensor locations affect vGRF estimates (Fig 2). A second innovation comes from our characterization of accelerometers to detect changes in vGRF across multiple loading phenotypes, prescribed herein using real-time biofeedback. As one clinical application, vGRF profiles during walking are indicative of patient-reported and biological outcomes among individuals during recovery from anterior cruciate ligament (ACL) reconstruction [19,20]. Thus, this study suggests that wearable sensors can be used to detect and monitor vGRF changes between overloading and typical walking, which could help improve clinical outcomes during rehabilitation.

Generally, our measured vGRFs were able to differentiate between control and overloading conditions and the Waist model replicated those findings. Interestingly, the Foot-Waist model

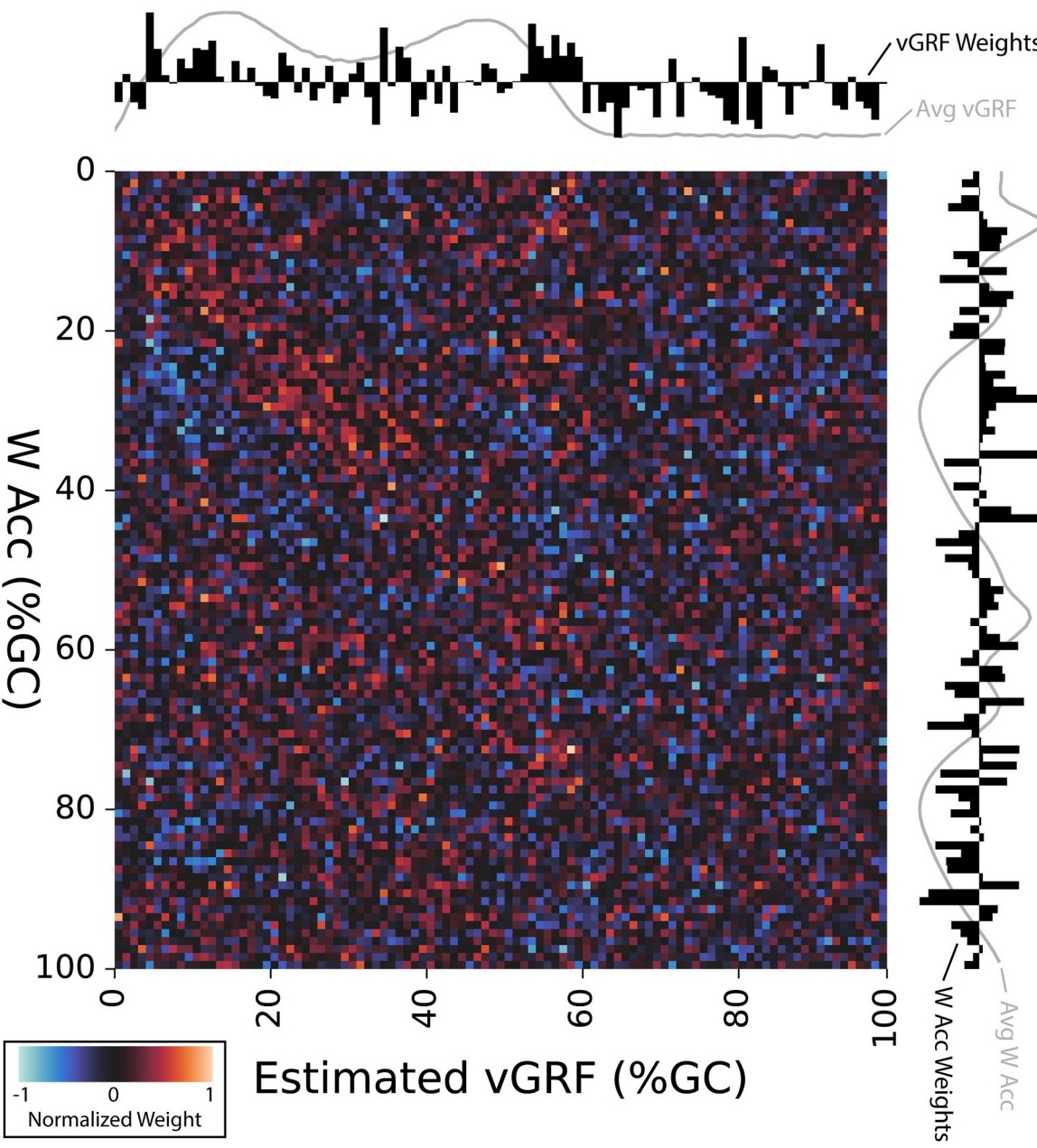

**Fig 6. To address interpretability of the Waist model, we show the resultant model coefficient weights for the input (waist acceleration or $W_{Acc}$, vertical axis) and output (vGRF, horizontal axis) layers of our multilayer perceptron neural network model.** Model weights are shown via heatmap as 1% instances of the gait cycle (%GC), along each axis from top-left to bottom-right, with higher positive weights in red, lower negative weights in blue, and zero weights in black. Along each axis, we show the mean-normalized summed weights for each column/row as a bar graph, with higher relative weights shown above or to the right of the axes. Because the $W_{Acc}$ is plotted along the vertical axis, the $W_{Acc}$ weights are shown to the right, aligned with the vertical axis. Similarly, the estimated vGRF is plotted along the horizontal axis of the heatmap, thus the vGRF weights are shown on top, in line with the horizontal axis. The black bars represent the mean-normalized neural net weight for that row ($W_{Acc}$, vertical axis) or column (Avg vGRF, horizontal axis) at that instance of the gait cycle. For additional context, we also overlay the average Waist acceleration vector magnitude (Avg $W_{Acc}$) along the vertical axis and the average estimated vGRF (Avg vGRF) along the horizontal sub-axis, both shown in gray.

detected a significant difference between underloading peak vGRF and the control condition that was not detected using the measured vGRF signals (Fig 5G). Although loading peaks did decrease on average during the underloading condition, no other model or measurement

mode detected a significant difference from the control vGRF. Loading peak errors were also higher on average during the underloading condition (Fig 5H), which may have contributed to this unanticipated finding. In combination with inaccuracy during overloading, we suggest that the Foot-Waist configuration is unlikely to yield accurate vGRF loading peak estimates across biofeedback conditions.

Our Waist model aligns with other groups that have used machine learning or regression equations to estimate vGRF loading peak during walking or running [22,23]. Although lower leg sensor sites can accurately estimate vGRF signals under different scenarios [24,25], we found that accelerometers placed near the body's center of mass may provide optimal features to estimate vGRF during walking. This outcome is theoretically supported by simple Newtonian physics-based models of walking [21], in that acceleration of the body's center of mass is proportional to the net external force acting on the body. While the other lower-limb sensors may contain salient features that can accurately estimate vGRF, we contend that sensor locations that track center of mass acceleration are more reliable and accurate.

We chose an accelerometer-based solution for vGRF estimation because of their low cost and simplicity, both of which are beneficial to delivering options for estimating vGRF outside of the research laboratory. Compared to other potential solutions (pressure-sensing insoles, instrumented walkways), accelerometers and inertial measurement units are adaptable across patients of any size, simple to change between patients, and can be applied to a wide variety of outpatient environments. Unlike previous studies, we used the acceleration vector magnitude in our model, combining the 3-dimensional components into one composite acceleration measure. This processing strategy simplifies sensor placement orientation, is agnostic to left/right sidedness, and reduces data processing complexity which should enable high-throughput assessment.

As scientists, engineers, and clinicians increasingly utilize machine learning models to estimate biomechanical parameters, interpretation of the data produced by these models remains a challenge for successful utilization of model-based techniques [28]. We assessed the interpretability of our most accurate model by mapping the input and output weights from the multilayer perceptron model (Fig 6). We found that model weights tended to align with notable events from the gait cycle, including stance and swing phase as well as local maximums in the vGRF signal during early and late stance. Our interpretability plot reveals some of the underlying features that our model may use to estimate the vGRF waveform. By calculating and displaying our model weights, we aim to work toward a better understanding of how machine learning models make predictions, and what input features are most influential, with the hope of understanding and explaining model estimates.

## Limitations

One limitation is that this study only includes young, healthy participants. Thus, our models may not extend to individuals with pathologic gait or who may have or be at risk for developing OA. We specifically included an underloading (-5%) condition in our protocol that emulates the less dynamic vGRF profiles (lower values in early and late stance, higher values during midstance, Fig 5A, 5E and 5I) commonly exhibited by individuals with knee OA [2,29,30]. This additional underloading condition serves as an early step towards generalizability in our models by including inputs similar to those from the target population. Nevertheless, future studies will be needed to include individuals with knee OA who typically differ from our control group by age, body mass, health status, and other demographics.

Another limitation is that our input data only included acceleration signals, lacking the gyroscope and magnetometer signals common to inertial measurement units. Although

accelerometers likely contain the most relevant features for vGRF estimation, it is possible that the other signals may contain salient features that could aid in vGRF estimation and perhaps help distinguish between loading conditions.

## Conclusion

We can accurately estimate the vGRF loading peak within 4.4% of body weight on average and distinguish between biofeedback-induced loading conditions in healthy participants using only a waist accelerometer by implementing a multilayer perceptron neural network model. However, other lower-body segment accelerations may be helpful in other ways, identifying gait events which are necessary to parse gait cycles from the acceleration signal stream. We suggest that a wearable sensor system comprised of a single waist sensor is not only sufficient, but optimal for implementing a biofeedback-based therapy system to prescribe limb loading during gait retraining in healthy participants.

## Author Contributions

**Conceptualization:** Ricky Pimentel, Cortney Armitano-Lago, Ryan MacPherson, Adam W. Kiefer, Brian Pietrosimone, Jason R. Franz.

**Data curation:** Ricky Pimentel, Cortney Armitano-Lago, Ryan MacPherson, Anoop Sathyan.

**Formal analysis:** Ricky Pimentel, Ryan MacPherson, Anoop Sathyan, Edgar Lobaton.

**Funding acquisition:** Michael Daniele, Adam W. Kiefer, Brian Pietrosimone, Jason R. Franz.

**Investigation:** Ricky Pimentel, Cortney Armitano-Lago, Ryan MacPherson, Adam W. Kiefer, Brian Pietrosimone, Jason R. Franz.

**Methodology:** Ricky Pimentel, Cortney Armitano-Lago, Ryan MacPherson, Anoop Sathyan, Jack Twiddy, Kaila Peterson, Adam W. Kiefer, Brian Pietrosimone, Jason R. Franz.

**Project administration:** Cortney Armitano-Lago, Brian Pietrosimone, Jason R. Franz.

**Resources:** Jack Twiddy, Michael Daniele, Adam W. Kiefer, Brian Pietrosimone, Jason R. Franz.

**Software:** Ricky Pimentel, Ryan MacPherson, Anoop Sathyan, Edgar Lobaton.

**Supervision:** Cortney Armitano-Lago, Michael Daniele, Adam W. Kiefer, Brian Pietrosimone, Jason R. Franz.

**Validation:** Ricky Pimentel, Cortney Armitano-Lago, Anoop Sathyan, Edgar Lobaton, Jason R. Franz.

**Visualization:** Ricky Pimentel, Anoop Sathyan, Jack Twiddy, Kaila Peterson, Edgar Lobaton, Jason R. Franz.

**Writing – original draft:** Ricky Pimentel.

**Writing – review & editing:** Ricky Pimentel, Jack Twiddy, Kaila Peterson, Adam W. Kiefer, Brian Pietrosimone, Jason R. Franz.

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
