## [Decision Letter · Decision Letter 0]

31 Oct 2023

PDIG-D-23-00298

Effect of sensor number and location on accelerometry-based vertical ground reaction force estimation during walking

PLOS Digital Health

Dear Dr. Franz,

Thank you for submitting your manuscript to PLOS Digital Health. After careful consideration, we feel that it has merit but does not fully meet PLOS Digital Health's publication criteria as it currently stands. Therefore, we invite you to submit a revised version of the manuscript that addresses the points raised during the review process.

Please submit your revised manuscript within 60 days Dec 30 2023 11:59PM. If you will need more time than this to complete your revisions, please reply to this message or contact the journal office at digitalhealth@plos.org. Please include the following items when submitting your revised manuscript:

We look forward to receiving your revised manuscript.

Kind regards,

Juan S. Osorio-Valencia, MSc MPH

Guest Editor

PLOS Digital Health

Journal Requirements:

1. Please send a completed 'Competing Interests' statement, including any COIs declared by your co-authors. If you have no competing interests to declare, please state "The authors have declared that no competing interests exist". Otherwise please declare all competing interests beginning with twhe statement "I have read the journal's policy and the authors of this manuscript have the following competing interests:"

Additional Editor Comments (if provided):

I find the study to be relevant and well-organized. However, I concur with our reviewers that there are areas requiring clarification, particularly in the methods section.

I would like to request a major revision, as suggested by our reviewers. Their comments to the authors contain specific questions that need to be addressed in the revision.

In my role as a Guest Editor for the journal, I'd like to highlight the importance of diversity in digital health research. If the study primarily involved young and healthy volunteers, it's crucial to discuss how the results might translate to patients with Osteoarthritis (OA), who typically differ in age, body weight, and health status. While you've mentioned this in the limitations, elaborating on the specific characteristics of patients with OA and how they may impact the model's performance is essential.

In addition to the changes recommended by our reviewers, please consider the following minor revisions:

Line 200: Referring to Fig. 5G, the hashtag (#) should be placed in the bar representing estimated values to demonstrate the comparison with measured values.

Line 206: Referring to Fig. 5K, the same applies here. The # should be positioned over the bar for measured values to signify significant differences with the measured values.

Line 264: You mentioned using the "acceleration vector magnitude" and combining the 3-dimensional components into one composite acceleration measure. Please ensure that this is clearly explained in the methodology.

Line 275: You discussed that the "interpretability plot reveals some of the underlying features that our model may use to estimate the vGRF." Please provide a clearer explanation of Fig. 6, both in the results section and the captions. Pay particular attention to the side graphs of mean-normalized summed weights.

I believe that addressing these points will significantly enhance the quality and clarity of your paper.

Reviewers' comments:

Reviewer's Responses to Questions

**Comments to the Author**

1. Does this manuscript meet PLOS Digital Health’s publication criteria? Is the manuscript technically sound, and do the data support the conclusions? The manuscript must describe methodologically and ethically rigorous research with conclusions that are appropriately drawn based on the data presented.

Reviewer #1: Yes

Reviewer #2: Yes

2. Has the statistical analysis been performed appropriately and rigorously?

Reviewer #1: Yes

Reviewer #2: Yes

3. Have the authors made all data underlying the findings in their manuscript fully available (please refer to the Data Availability Statement at the start of the manuscript PDF file)?

Reviewer #1: Yes

Reviewer #2: Yes

4. Is the manuscript presented in an intelligible fashion and written in standard English?

Reviewer #1: Yes

Reviewer #2: Yes

5. Review Comments to the Author

Reviewer #1: The work is interesting and give some important information about the vGRF. However I have some questions regarding the methods and the way the authors take the data, because is not clear in the paper. What is the biofeedback used, how is measured (I guess using the treadmill but is not clear)? 

Is the information in the screen part of the bio feedback? 

The EMG sensors (Delsys Trigno) are used to collect information, but later is not clear how this information is used to compare with accelerometers. 

Because the model is not mentioned, is not clear if the sensors include EMG or just are IMU (Not only accelerometers).

 It seems that acelerometers are only compared with the treadmill information.

Bassically, the methods section need to be improved, in order to better understand the work and its results. 

The training of the neural networks used needs to be explained, because is not clear the relation between the trained model using the treadmill and the accelerometer. Are the same or only this training was used with the acelerometers?

Reviewer #2: This is a well conducted study that extended previous work on using wearable sensors to estimate ground reaction force. The novelty of this study lies in the systematic investigation of various sensor configurations on the estimation of ground reaction force during walking. The manuscript was well written. The methods are appropriate overall, but I have one comment on the methods: 

1. I understand that both the left and right GRFs and accelerations (50 strides from each side) were combined together for training and testing of machine learning model. Have the authors looked at the analysis using only one side of data to build machine learning model for each side separately? Would it be possible that this way of analysis will show that the other configuration of sensors might actually perform better than waist only model? This is because that there might be asymmetry in steps of the healthy participants, which makes the waist model being advantageous than other configurations overall. However, this advantage might change if data from one side was used separately, especially in participants with larger asymmetry. It might also be worth checking using statistics whether asymmetry had any effect on the error of the models. This may be relevant to the clinical application of the model as asymmetry is often observed on patients with OA.

6. PLOS authors have the option to publish the peer review history of their article (what does this mean?). If published, this will include your full peer review and any attached files.

**Do you want your identity to be public for this peer review?** For information about this choice, including consent withdrawal, please see our Privacy Policy.

Reviewer #1: No

Reviewer #2: No

---

## [Decision Letter · Decision Letter 1]

5 Mar 2024

PDIG-D-23-00298R1

Effect of sensor number and location on accelerometry-based vertical ground reaction force estimation during walking

PLOS Digital Health

Dear Dr. Franz,

Thank you for submitting your manuscript to PLOS Digital Health. After careful consideration, we feel that it has merit but does not fully meet PLOS Digital Health's publication criteria as it currently stands. Therefore, we invite you to submit a revised version of the manuscript that addresses the points raised during the review process.

Please submit your revised manuscript within 30 days Apr 04 2024 11:59PM. If you will need more time than this to complete your revisions, please reply to this message or contact the journal office at digitalhealth@plos.org. Please include the following items when submitting your revised manuscript:

We look forward to receiving your revised manuscript.

Kind regards,

Juan S. Osorio-Valencia, MSc

Guest Editor

PLOS Digital Health

Journal Requirements:

Additional Editor Comments (if provided):

The paper has improved substantially. Please address the minor comments provided by one of the reviewers. Thank you.

Reviewers' comments:

Reviewer's Responses to Questions

**Comments to the Author**

1. If the authors have adequately addressed your comments raised in a previous round of review and you feel that this manuscript is now acceptable for publication, you may indicate that here to bypass the “Comments to the Author” section, enter your conflict of interest statement in the “Confidential to Editor” section, and submit your "Accept" recommendation.

Reviewer #1: All comments have been addressed

Reviewer #2: (No Response)

2. Does this manuscript meet PLOS Digital Health’s publication criteria? Is the manuscript technically sound, and do the data support the conclusions? The manuscript must describe methodologically and ethically rigorous research with conclusions that are appropriately drawn based on the data presented.

Reviewer #1: Yes

Reviewer #2: Yes

3. Has the statistical analysis been performed appropriately and rigorously?

Reviewer #1: Yes

Reviewer #2: Yes

4. Have the authors made all data underlying the findings in their manuscript fully available (please refer to the Data Availability Statement at the start of the manuscript PDF file)?

Reviewer #1: Yes

Reviewer #2: Yes

5. Is the manuscript presented in an intelligible fashion and written in standard English?

Reviewer #1: Yes

Reviewer #2: Yes

6. Review Comments to the Author

Reviewer #1: Authors have addressed all the comments and corrected the issues. The paper is highly improved and very clear now. The contribution is important.

Reviewer #2: Thanks for authors’ response. It is clear from the provided data that there was no asymmetry in the study sample. Therefore, I would agree with the authors that separating left and right might not improve the accuracy of the model. However, I would suggest make this clear in the conclusion that this model is only for health participants with little or no asymmetry. Please see specific suggestions below:

Page 2, line 39. Abstract: the last sentence may be written as “Our results suggest that a system designed to monitor changes in walking vGRF or to deploy targeted biofeedback may only need a single accelerometer located at the waist for healthy participants.”

Page 3 line 55: Author summary: the last sentence may be written as “Our results suggest that a system designed to monitor changes in walking vGRF or to deploy targeted biofeedback may only need a single accelerometer located at the waist for healthy participants.”

Page 11 line 246: Discussion: the last sentence may be written as “Our results suggest that a system designed to monitor changes in walking vGRF or to deploy targeted biofeedback may only need a single accelerometer located at the waist for healthy participants.”

Page 14 line 309-315: the conclusion may be written as “We can accurately estimate the vGRF loading peak within 5% of body weight and distinguish between biofeedback-induced loading conditions in healthy participants using only a waist accelerometer by implementing a multilayer perceptron neural network model. ….. We suggest that a wearable sensor system comprised of a single waist sensor is not only sufficient, but optimal for implementing a biofeedback-based therapy system to prescribe limb loading during gait retraining in healthy participants.”

7. PLOS authors have the option to publish the peer review history of their article (what does this mean?). If published, this will include your full peer review and any attached files.

**Do you want your identity to be public for this peer review?** For information about this choice, including consent withdrawal, please see our Privacy Policy. 

Reviewer #1: No

Reviewer #2: No

---

## [Editor Report · Decision Letter 2]

5 Apr 2024

Effect of sensor number and location on accelerometry-based vertical ground reaction force estimation during walking

PDIG-D-23-00298R2

Dear Dr. Franz,

We are pleased to inform you that your manuscript 'Effect of sensor number and location on accelerometry-based vertical ground reaction force estimation during walking' has been provisionally accepted for publication in PLOS Digital Health.

Best regards,

Juan S. Osorio-Valencia, MSc

Guest Editor

PLOS Digital Health

Reviewer Comments (if any, and for reference): 

I would like to express my gratitude to the entire team for their diligent efforts in preparing the article for publication. Your hard work and dedication have ensured its suitability for the journal. Congratulations on this achievement! I particularly commend the importance it has for implementing a biofeedback-based therapy system in knee osteoarthritis. Once again, well done, and thank you for your contributions.